# Profile of Newly Diagnosed Patients with HIV Infection in North-Eastern Romania

**DOI:** 10.3390/medicina59030440

**Published:** 2023-02-23

**Authors:** Isabela Ioana Loghin, Andrei Vâţă, Ioana Florina Mihai, George Silvaş, Şerban Alin Rusu, Cătălina Mihaela Luca, Carmen Mihaela Dorobăţ

**Affiliations:** 1Department of Infectious Diseases, “Grigore T. Popa” University of Medicine and Pharmacy, 700115 Iasi, Romania; 2Department of Infectious Diseases, “St. Parascheva” Clinical Hospital of Infectious Diseases, 700116 Iasi, Romania

**Keywords:** HIV/AIDS, opportunistic infections, HBV, HCV, ART, UNAIDS

## Abstract

*Background and Objectives*: Human immunodeficiency virus infection and the acquired immunodeficiency syndrome (HIV/AIDS) pandemic are unquestionably the most serious public crisis of our time. Identifying, preventing, and treating HIV-associated comorbidities remains a challenge that must be addressed even in the era of antiretroviral therapy. *Materials and Methods*: In this study, we aimed to characterize the aspects of newly diagnosed patients with HIV/AIDS, during 2021–2022 in Northeastern Romania. We reviewed the frequency and associated comorbidities of these patients in correspondence with national and global results. *Results*: Our study found that of all newly diagnosed HIV cases (167 cases—74 cases in 2021 and 98 cases in 2022), 49.70% were diagnosed with HIV infection and 50.30% had AIDS. Based on sex correlated with the CD4+ T-lymphocyte level, the most affected were males, with a lower CD4+ T-lymphocyte level overall. The average HIV viral load was 944,689.55 copies/mL. Half of males had an abnormal ALT or AST (39.53% and 49.61%); as for the females, less than a quarter had an increased value of ALT or AST, respectively (18% and 26%). The most frequent co-infections were as follows: oral candidiasis (34.73% of patients), hepatitis B (17.37% of patients), and SARS-CoV-2 infection (8.38%), followed by hepatitis C (6.39%), tuberculosis (TB), syphilis, toxoplasmosis, *Cryptococcus*, *Cytomegalovirus* infections. Males were more affected than females, with a higher percentage of co-infections. The prescribed antiretroviral treatment focused on a single-pill regimen (79.04%) to ensure adherence, effectiveness, and safety. Therefore, 20.96% had been prescribed a regimen according to their comorbidities. *Conclusions*: Our study found a concerning rise in the incidence of HIV in 2022 compared to that in 2021 in Northeastern Romania, because of the rise in post-SARS-CoV-2 pandemic addressability. Advanced immunodeficiency and the burden of opportunistic infections characterize newly diagnosed HIV patients. The physicians should keep in mind that these patients may have more than one clinical condition at presentation.

## 1. Introduction

Human immunodeficiency virus infection and the acquired immunodeficiency syndrome (HIV/AIDS) pandemic are unquestionably the most serious public crisis of our time despite the international and local efforts to combat this calamity. The life expectancy of HIV-positive individuals who receive efficient antiretroviral therapy (ART) and sustained viral suppression is close to normal. Identifying, preventing, and treating HIV-associated comorbidities remains a challenge that must be addressed even in the era of antiretroviral therapy.

Patients who have recently been diagnosed with HIV infection may be asymptomatic or develop a wide range of symptoms associated with opportunistic infections, acute seroconversion disease, or other disorders [1].

HIV infection can spread through three different routes: parenteral, vertical, and sexual. Among women, heterosexual transmission has decreased substantially in Europe in recent years. While transmission through injected drug use has declined steadily since 2012, it remains high in the East [2].

In addition to traditional risk factors, such as age, dyslipidemia, diabetes, lipodystrophy, high blood pressure, obesity, smoking, and drug use, comorbidities may be caused by HIV infection itself (through microbial translocation, low-grade persistent chronic inflammation, immune system activation, and pro-coagulant mechanisms), co-infections (with hepatitis viruses, herpes viruses), opportunistic infections (*Myco-bacterium tuberculosis, Cryptococcus neoformans, Pneumocystis jiroveci)* and ART (toxicities, drug–drug interactions) [3]. The Joint United Nations Programme on HIV/AIDS (UNAIDS) confirmed that 38.4 million people globally were infected with HIV in 2021 from which 1.5 million people became newly infected with HIV in the year 2021 and 650,000 people died from AIDS-related illnesses in 2021.

New HIV infections have been reduced by 54% since their peak in 1996. In 2021, around 1.5 million people were newly infected with HIV, compared to 3.2 million people in 1996. Women and girls accounted for 49% of all new infections in 2021. Since 2010, new HIV infections have declined by 32%, from 2.2 million to 1.5 million. Since 2010, new HIV infections among children have declined by 52%, from 320,000 in 2010 to 160,000 in 2021.

On World AIDS Day 2014, UNAIDS set targets aimed at ending the AIDS epidemic by 2030. To achieve this, countries are working toward reaching the interim “95-95-95” targets of 95% of people living with HIV knowing their HIV status, 95% of people who know their HIV positive status based on treatment, and 95% of people on treatment with suppressed viral loads—by 2025. Of all people living with HIV, 85% knew their status, 75% were accessing treatment, and 68% were virally suppressed in 2021 [4].

In Romania, The National Institute of Infectious Diseases “Prof. Dr. Matei Balș” published statistics stating that since the beginning of 2022 up until the end of September 2022, 445 new HIV–AIDS patients were identified, from which 126 had passed away due to HIV–AIDS-associated conditions [3]. From 1985 to 2022, Romania recorded 26,791 HIV-infected cases, from which there were 10,053 pediatric cases, 16,738 adult cases, and 8293 patients passed away during this period [5].

In this study, we aimed to characterize the aspects of newly diagnosed patients with HIV/AIDS, during 2021–2022 in Northeastern Romania. We reviewed the frequency and associated comorbidities of the newly diagnosed patients with HIV infection in correspondence with national and global results.

## 2. Materials and Methods

### 2.1. Database Description

We conducted a retrospective clinical study, based on hospital medical records of newly diagnosed patients with HIV/AIDS, in Northeastern Romania, hospitalized in the “Sf. Parascheva” Clinical Hospital of Infectious Diseases from Iasi, aiming to highlight the profile and associated comorbidities of the new HIV/AIDS cases, in the context of the SARS CoV-2 pandemic. The studied period was between 1 January 2021 and 31 December 2022.

Inclusion criteria selected patients over 18-years-old with an HIV-positive enzyme-linked immunosorbent assay (ELISA) test and confirmation of HIV/AIDS infection via Western blotting (WB), hospitalized in our Regional HIV/AIDS Center of Northeastern Romania. Patients who tested positive for HIV/AIDS infection were also evaluated based on the HIV plasma viral load and CD4+ T cell levels. Our study group included 74 patients in 2021 and 93 patients in 2022.

The study obtained the approval of the Ethics Committee of the “Sf. Parascheva” Clinical Hospital of Infectious Diseases, Iasi, Romania. (Approval No. 32/5 December 2022). All participants signed informed consent at the time of admission.

The collected data included demographic aspects (age, sex), personal pathological antecedents, clinical characteristics, blood tests (viro-immunological testing), assessment of potential associated opportunistic infections, patient staging, antiretroviral treatment initiated, and the evolution and prognosis after therapy of patients newly diagnosed with HIV/AIDS infection.

The HIV infection stage, based on age-specific CD4+ T-lymphocyte counts or CD4+ T-lymphocyte percentage of total lymphocytes CD4 T cells level, was established according to the Centers for Disease Control and Prevention (CDC) Atlanta Classification—surveillance data on HIV infection and AIDS: stage 1, when the CD4+ T-lymphocyte level is ≥500 cells/μL; stage 2, CD4+ T-lymphocytes between 200 and 499 cells/μL; and stage 3, a CD4+ T-lymphocyte level ≤200 cells/μL. Stages 1 and 2 represent HIV infection and stage 3 is associated with AIDS [4,6].

The people suspected of having HIV were serologically evaluated through two ELISA tests, and the serological confirmation was achieved by performing the Western blot test. All this was carried out by the epidemiologists within the regional public health management network, and later, the patients were directed to the regional HIV/AIDS center.

All blood tests were performed by the hospital’s central laboratory, and the HIV plasmatic viral load and CD4+ T cell level were assessed by the hospital’s molecular biology laboratory. The method used for identifying HIV viremia, as well as monitoring the viral load levels, was a measurement based on RT-PCR HIV 1 using Cepheid’s GeneXpert^®^ (Headquarters: Sunnyvale, California, U.S, Factory: Solna, Sweden). The viral load was considered undetectable when values were under 40 copies/mL and detectable when values were above 40 copies/mL.

All newly diagnosed PWH (people with HIV) were clinically and biologically evaluated periodically, for metabolic syndrome and liver enzymes. The laboratory reference values were between 5 and 31 UI/l for ALT (alanine transaminase) and for AST (aspartate transaminase), between 7 and 32 UI/l for GGT (gamma-glutamyl transferase), between 122 and 200 mg/dL for COL (cholesterol), between 40 and 66 mg/dL for HDL-COL (high-density lipoprotein cholesterol), 30 and 159 mg/dL for LDL-COL (low-density lipoprotein cholesterol), and 30 and 150 mg/dL for TG (triglycerides), with no differences between sexes.

### 2.2. Statistical Analysis

The correlation analysis among demographic parameters, clinical data, and outcomes was performed using the Pearson test in XLSTAT version 2019 software (ADDINSOFT, Paris, France) Kendall’s Τau correlation coefficients were calculated (11). Statistical analysis was performed using Statistical Software for Excel (XLSTAT) version 2019 (Addinsoft, New York, NY, USA).

## 3. Results

In the Northeastern part of Romania, in 2022 and 2021, there was a total of 167 cases (93 in 2022, representing 55.69%, and 74 cases in 2021, representing 44.31%), and HIV infection was most frequent in men (129 cases, 77.25%) than in women (38 cases, 22.75%) (Figure 1).

The majority of cases were young adults, aged between 31- and 40-years-old—67 patients (40.12%), followed by the age group 21–30—48 patients (28.74%), 41–50 years—22 patients (13.17%), 51–60-years-old—11 patients (6.59%), over 61-years-old—10 patients (6.99%), and 0–20 years—9 patients (5.39%) (Table 1). The average age in the study group was 35-years-old.

The distribution of our study group based on county showed that almost a third of the patients were from Iasi (48 cases, 28.74%), followed by Bacau (35 cases, 20.96%), Neamt (31 cases, 18.56%), Suceava (31 cases, 18.56%), Botosani (16 cases, 9.58%), and Vaslui (6 cases, 3.59%) (Table 2). From the urban area in Northeast Romania, there were 101 patients (60.48%), and the remaining 66 cases (39.52%) were from rural areas (Figure 2).

Considering the route of transmission, only 159 cases (95.21%) reported a possible cause. For the remaining eight cases (4.79%), the route of transmission remains unknown (Table 3).

Regarding the sexual route of transmission (heterosexual and MSM/men having sex with men—90.42% cases), the most affected group was young adult males (aged 21–40) with a medium education level. Intravenous drug use was recorded at 2.99% with three perinatal cases (1.8%).

All of the newly diagnosed patients from the Iasi HIV/AIDS Regional Center between 1 January 2021 and 31 December 2022 were virologically and immunologically evaluated.

It was observed that 43.11% of cases had a CD4+ T-lymphocyte level between 1 and 199 cells/μL, 37.72% of cases had a CD4+ T-lymphocyte value between 200 and 499 cells/μL, and 19.16% had CD4+ T-lymphocyte values over 500 cells/μL, with an average CD4+ T-lymphocyte level of 300.45 cells/μL (Table 4, Figure 3 and Figure 4).

Based on gender correlated with the CD4+ T-lymphocyte level, the most affected were males, with a lower CD4+ T-lymphocyte level overall. The average HIV viral load was 944,689.55 copies/mL.

We used the CDC (Center for Disease Control and Prevention) stages of HIV/AIDS, and the results showed the following: 32 patients were stage 1 HIV infection (19.16%), 51 patients were in the 2nd stage (30.54%), and 84 patients in the 3rd stage (50.3%) (Table 5).

In the study group, it was observed that almost half of the males had an abnormal ALT or AST (39.53% and 49.61%); as for the females, less than a quarter had an increased value of ALT or AST, respectively (18% and 26%). Of 39.53% of the males with an abnormal ALT value, 5.43% (7 cases) had hepatitis B, 0.78% (1 case) had hepatitis C, and only 9.3% (12 cases) declared occasional alcohol consumption. Of 49.61% of male patients with abnormal AST, 3.1% (4 cases) had hepatitis B, 1.55% (2 cases) had hepatitis C, and only 12.48% (16 cases) declared occasional alcohol consumption. The rest of the patients did not have an identified cause of elevated transaminase values.

Regarding the metabolic profile, cholesterol levels were increased in a third of the study group, regardless of sex (31.01% males and 34% females); the triglyceride levels were more than a third, affecting almost equally both sexes (44.19% males and 39% females) (Table 6).

The study group was screened for the most common co-infections associated with HIV/AIDS. The results showed that two-thirds (65.87%) of the patients admitted to our clinic in the studied period had different opportunistic infections. Many of the opportunistic infections were identified in stages 2 and 3 (25.75%, and 32.34%), when the CD4 T-lymphocyte level was under 500 cells/μL (Table 7).

The results showed that the most frequent co-infections were oral candidiasis (34.73% of patients), hepatitis B (17.37% of patients), and SARS-CoV-2 infection (8.38%), followed by hepatitis C (6.39%). To a lesser extent, cases of tuberculosis (TB), syphilis, toxoplasmosis, *Cryptococcus*, *Cytomegalic virus* (CMV), *Herpes virus*, and *varicella-zoster virus* (VZV) infections were recorded. Males were more affected than females, with a higher percentage of co-infections (Table 8).

All of the newly diagnosed patients in the Iasi HIV/AIDS Regional Center were prescribed ART (antiretroviral treatment) with a focus on a single-pill regimen to ensure adherence to treatment. Therefore, 79.04% of cases were prescribed a single-pill regimen, and the remaining 20.96% had been prescribed a regimen that took into account their comorbidities (Table 9). The treatment was initiated as soon as the diagnosis and antiretroviral therapy were established. The period of time between diagnosis and ART initiation ranged between 72 h and 14 days according to the severity of the cases, in order to avoid IRIS (immune reconstruction inflammatory syndrome), also following www.hiv-druginteractions.org.

The patients were evaluated after one month, and the viro-immunological status showed, in Table 9, an increased CD4+ T-lymphocyte level and a significant decrease in HIV viremia. As such, 56 patients (33.53%) had a CD4 value between 1 and 199 cells/μL, 74 patients (44.31%) had a value between 200 and 499 cells/μL, and 37 patients (22.16%) had a value above 500 cells/μL. The median CD4 value was 363.52 cells/μL. The average HIV viral load was 57,907.2 copies/mL. In both sexes, most of the cases had a CD4+ T-lymphocyte level between 200 and 499 cells/μL (Table 10, Figure 5).

The patients were initially evaluated at diagnosis and then one month after starting ART. The HIV viral load showed a significant decrease after starting antiretroviral therapy, with viral suppression being obtained in 50.9% of cases (85 cases) (Table 11).

## 4. Discussion

HIV infection is becoming a chronic disease because of effective ART treatment regimens. More HIV-infected patients are developing additional chronic diseases due to an increase in the quality of life, which is becoming close to normal. HIV-related healthcare requirements will rise, putting more strain on health systems and significantly impacting public health.

Our study found that of all the newly diagnosed HIV cases (167 cases), 49.70% were diagnosed with HIV infection and 50.30% had AIDS-related symptoms. Moreover, we found a concerning rise in the incidence of HIV in 2022 (93 cases) compared to that in 2021 (74 cases) in our region (North-eastern Romania), probably because of the rise in post-SARS-CoV-2 pandemic addressability.

The number of new HIV diagnoses in 2021 in Europe was 16,624 cases, with 40% (6648 cases) of all new HIV diagnoses in 2021 and more than half (55%) of diagnoses with the route of transmission known. MSM continues to be the most common method of HIV transmission recorded in the European Union. More than 60% of new HIV diagnoses, among those with a documented route of HIV transmission, were caused by sexual contact between men. In Europe, heterosexual contact accounted for 29% (4848 cases) of HIV infections, and in 40% of the number of cases, the mode of transmission was known, making it the second most often reported mode of HIV transmission. Nearly 4% of HIV diagnoses in 2021 were attributable to the transmission through injected drugs. Less than 1% of new HIV diagnoses in Europe in 2021 were attributable to vertical transmission, while 27% of new HIV diagnoses did not have a known mode of transmission. In East Europe, in 2020, HIV infection in new cases was transmitted through sex between men 3.1%, heterosexual transmission (men) at 32.7%, heterosexual transmission (women) at 32.6%, injected drug use at 28.1%, mother-to-child transmission at 28.1%, and unknown at 3.0%. In our region, we had similar results; from 159 cases, the most frequent was sexual transmission (heterosexual contact and MSM) (90.42%), with only a few known cases of intravenous drug usage and perinatal cases.

In the WHO European Region, 43 countries reported 8.194 new AIDS cases in 2021, with a diagnosis rate of 1.2 per 100,000 people. Over the past ten years, the number of AIDS cases has steadily decreased in the West, European Union, and the East. Furthermore, in the East, the number of infections stabilized between 2012 and 2018 and even decreased in 2019. The rate continued to fall in 2020–2021, though it is possible that this was because of the COVID-19 pandemic’s impact on reporting [2]. In our region, we also found a decreasing rate; in 2021, we had admitted, from a total of 159 cases, 74 patients (44.31%), and in 2022, 93 patients (55.69%).

In a study from South Asia, it was found that the mean age of HIV/AIDS patients was 36.38 ± 10.62 years. The data presented are consistent with those from our study (mean age of our study group—35.35). The most common symptoms were fever (28.89%), weight loss (28.61%), and generalized weakness (22.22%). The overall mean CD4 count was 176.04 ± 163.49 cells/mm^3^. Regarding the results from our study group, the mean CD4 count was 300.45 cells/mm^3^. There were 224 opportunistic infections documented in 160 patients, with opportunistic diarrhea (12.22%) and pulmonary tuberculosis (10.83%) being the most common. In our study group, opportunistic infections were documented in two-thirds of the patients (65.87%). The majority of the patients (80.83%) were eligible for the initiation of first-line antiretrovirals at presentation, while all patients from our clinic initiated antiretroviral therapy [7].

In a cohort from Morocco, they treated 525 patients with new HIV diagnoses during the course of 18 months. The sex ratio was 1:1, and the mean age was 36.1 years. While the sex ratio in our study was 3:1, the mean age had similar results (35.35). In 47.8% of instances, the seropositivity was identified based on an evocative symptom. The two primary clinical signs were oral candidiasis (11.2%) and weight loss (16.6%). The majority (23%) of opportunistic infections were tuberculosis. A stage of acquired immunodeficiency syndrome (AIDS) diagnosis was made in 36.1% of cases. In our research, we found that almost half of the patients newly diagnosed were in the AIDS stage. The first CD4 count and viral load tested had median values of 248/mm^3^ and 88.174 copies/mL, respectively, results comparable with our paper (median CD4 count, 300.45 cells/mm^3^), but the median viral load was increasingly higher (944,689.55 copies/mL) [8].

Gokengin D et. al. realized a survey about HIV care, which included twenty-four countries (Albania, Armenia, Azerbaijan, Bosnia and Herzegovina, Bulgaria, Croatia, Czech Republic, Estonia, Georgia, Hungary, Kazakhstan, Kosovo, Kyrgyz Republic, FYR of Macedonia, Moldova, Montenegro, Poland, Romania, Russian Federation, Serbia, Slovak Republic, Slovenia, Turkey, and Uzbekistan) out of 31 (77.4%) from Central and Eastern Europe. The major route of transmission was MSM (41.7%, 10/24 countries), followed by heterosexual contact (37.5%, 9/24 countries) and injected drug use (20.8%, 5/24 countries). Men who have sex with men (MSM) (14/24, 58.3%), persons who inject drugs (15/24, 62.5%), and sex workers (12/24, 50.0%) were among the other categories subjected to targeted screening. Pregnant women were only screened in 14 of the 24 countries (58.3%) [9]. The frequency in transmission routes was similar to that in our region.

In a Spanish study, from a total of 1398 HIV-positive individuals, 2.1% of the infections involved injected drugs or slam practices, and 97.9% of the infections were sexually transmitted. The median age was 32.9 years, comparable to our results, and 40.1% of the population was Latin American [10].

In a study from France about comorbidities in people living with HIV in comparison to those in non-HIV people, the researchers showed that PWH had significantly higher rates of alcohol abuse (5.8% vs. 3.1%), chronic renal disease (1.2% vs. 0.3%), cardiovascular disease (7.4% vs. 5.1%), dyslipidemia (22% vs. 15.9%), and hepatitis B (3.8% vs. 0.1%) [11]. Moreover, PWH had higher rates of other comorbidities, including anemia, malnutrition, mental disorders, and tumors [12].

Other researchers observed that the most common non-related HIV-comorbidities were vitamin D deficiency (29.1%), depressive episodes (27.8%), arterial hypertension (16.3%), and hypercholesterolemia (10.8%) in people with HIV in Germany [13]. Another German study found that the prevalence in PWH compared to that in the matched non-HIV cohort of acute renal disease (0.5% vs. 0.2%), bone fractures due to osteoporosis (6.4% vs. 2.1%), chronic renal disease (4.3% vs. 2.4%), cardiovascular disease (12.8% vs. 10.4%), Hepatitis B (5.9% vs. 0.3%), and Hepatitis C infection (8.8% vs. 0.3%) was significantly higher in PWH [14]. Our study showed that the most frequent co-infections/comorbidities were oral candidiasis (34.73%), hepatitis B (17.37%), SARS-CoV-2 infection (8.38%), and hepatitis C (6.39%).

Roomaney et al. found that cardiovascular diseases were more frequent in people with HIV (especially hypertension: 13.3%), and the next prevalent comorbidities were pulmonary diseases (tuberculosis was the main cause: 3.5%), followed by metabolic diseases, such as diabetes (3.0%) and cancer (0.4%). Elderly people were more likely to contract any of the diseases. In general, the prevalence of diseases, such as cancer, diabetes, heart disease, and hypertension, was higher in women [15].

Another recent study observed that HIV infections were seen more in males than in females, a fact that our study showed as well. A significant decrease in the complete blood count was observed in HIV patients when compared to that in healthy individuals. A significant increase in aspartate aminotransferase (AST), alanine aminotransferase (ALT), urea, and creatinine was observed in HIV patients, with results comparable with ours. No significant difference was observed in alkaline phosphatase (ALP), total bilirubin, and albumin levels when compared to those in the healthy controls. Anemia was observed in 59.4% of HIV patients. A total of three (8.1%) patients were found to be co-infected with hepatitis B and one (2.7%) was co-infected with hepatitis C. Our study aimed to correlate hepatitis B or C co-infection in HIV/AIDS patients with comparable results [16,17].

Harklerode et al. found that of the 8664 newly discovered HIV cases throughout the trial period on the territory of Kenia, 3.1% had an HIV retest after the first diagnosis. About half (45.3%) had links to care documented. The median CD4 count at baseline was 332 cells/mm^3^. Our study group’s mean CD4 count was 300.45 cells/mm^3^, and 53.0% of those newly diagnosed with HIV who had received a CD4 test and were 15 years old or older did so at a late stage, with 32.9% already having advanced HIV. Being male and older than 34 years of age were two characteristics linked to a late diagnosis [18].

A clustering of common comorbidities, such as cardiovascular diseases, metabolic disorders, sexually transmitted diseases (STDs), and mental health issues, was identified in two different cohorts of PWH [19,20], further extending our knowledge of comorbidity profiles in HIV. Importantly, it was revealed that these illnesses did not occur together randomly [21,22]. Our evaluation focused on the most common coinfections, although the study group was screened periodically for other possible illnesses. In our cohort of patients, the most frequent co-infections were oral candidiasis (34.73%), hepatitis B (17.37%), SARS-CoV-2 infection (8.38%), and hepatitis C (6.39%).

Among PWH from middle- and low-income countries, it was found that the summary risk for oral candidiasis, tuberculosis, herpes zoster, and bacterial pneumonia was highest (>5%) among ART-naive patients. With the exception of tuberculosis, all opportunistic infections experienced a reduction in incidence over the first year of ART (range: 57–91%). The reductions were greatest for oral candidiasis, *Pneumocystis* pneumonia, and toxoplasmosis [23,24]. To a lesser extent, cases of tuberculosis (TB) (5.39%), syphilis (1.80%), toxoplasmosis (1.80%), *Cryptococcus* (0.60%), *Cytomegalic virus* (CMV) (1.20%), *Herpes virus*, and *varicella-zoster virus* (VZV) (2.40%) infections were recorded. The favorable evolution after ART was described in Romania, which used a single-pill regimen, also adapted to patient’s comorbidities according to www.hiv-druginteractions.org [25,26].

Our results can be comparable to those of other studies, but on a smaller scale, given the size of the study group [27,28]. To create the most efficient healthcare programs possible, especially with the aging population, the medical multidisciplinary team must provide the patients with the resources they need to inform themselves about their disease and effectively manage their illness [29,30].

## 5. Conclusions

Our study found a concerning rise in the incidence of HIV in 2022 compared to that in 2021 in our region, probably because of the rise in post-SARS-COV-2 pandemic addressability.

Advanced immunodeficiency and the burden of opportunistic infections characterize newly diagnosed HIV patients. The physicians should keep in mind that these patients may have more than one clinical condition at presentation.

HIV infection is still detected after a long period in our country, despite decent antiretroviral coverage. The encouragement of voluntary testing, particularly among those at a high risk of infection (medical personnel, men having sex with men, people who inject drugs, youth under 24-years-old, sex workers, and people who are economically disadvantaged), must be the focus.

Additionally, we need to make HIV testing more accessible, with community outreach efforts being especially helpful. These outreach efforts should always be supported by assisted counseling and prompt referral to medical facilities to begin ART.

## Figures and Tables

**Figure 1 medicina-59-00440-f001:**
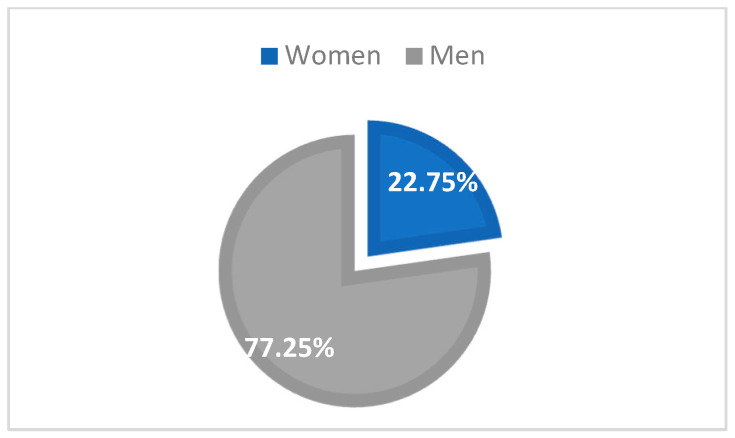
Distribution of new HIV/AIDS cases by sex.

**Figure 2 medicina-59-00440-f002:**
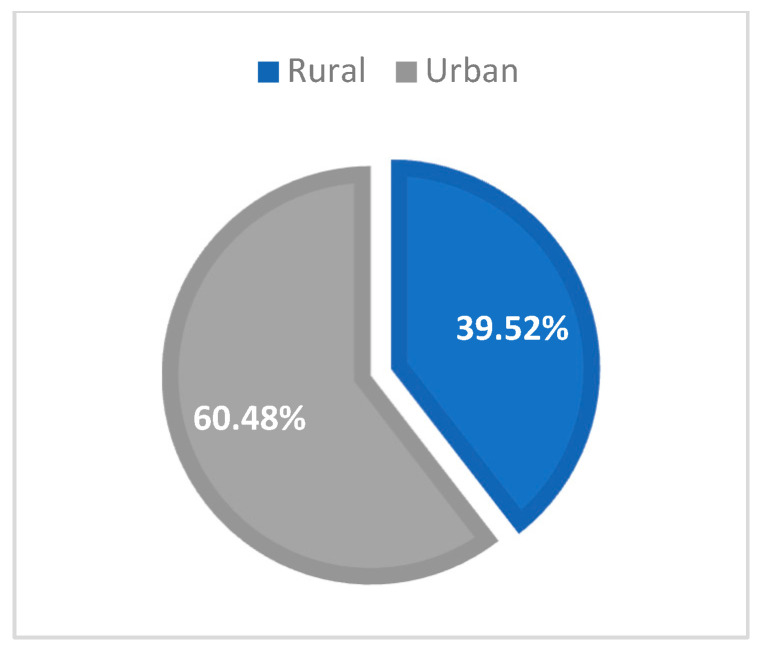
New HIV/AIDS cases, distribution by area.

**Figure 3 medicina-59-00440-f003:**
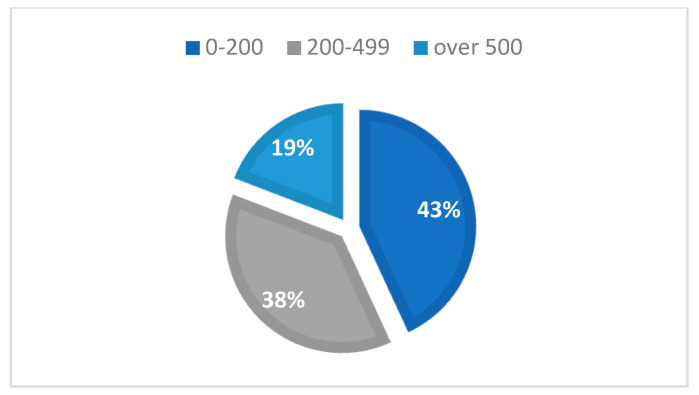
Distribution of cases by CD4+ T-lymphocyte level (cells/μL).

**Figure 4 medicina-59-00440-f004:**
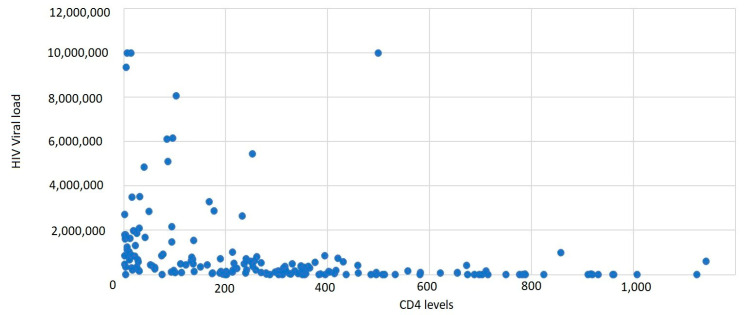
Distribution of new HIV/AIDS cases by CD4+ T-lymphocyte level and HIV viral load.

**Figure 5 medicina-59-00440-f005:**
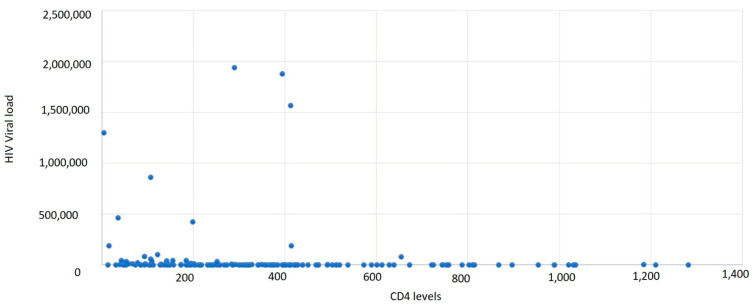
Distribution of HIV/AIDS cases by CD4+ T-lymphocytes level and HIV viral load, one month after ART.

**Table 1 medicina-59-00440-t001:** Distribution of new HIV/AIDS cases by age.

Age (Years)	*n*	%
0–20	9	5.39
21–30	48	28.74
31–40	67	40.12
41–50	22	13.17
51–60	11	6.59
over 61	10	5.99

**Table 2 medicina-59-00440-t002:** Distribution of new HIV/AIDS cases by county in Northeast Romania.

County	*n*	%
Iasi	48	28.74
Neamt	31	18.56
Vaslui	6	3.59
Bacau	35	20.96
Botosani	16	9.58
Suceava	31	18.56

**Table 3 medicina-59-00440-t003:** Route of transmission of the study group.

	*n*	%
Route of transmission	159	95.21
Sexual (heterosexual and MSM)	151	90.42
Intravenous drug-use	5	2.99
Perinatal	3	1.8
Unknown	8	4.79

**Table 4 medicina-59-00440-t004:** Distribution of new HIV/AIDS cases by CD4+ T-lymphocyte level and sex.

CD4+ T-lymphocyte Level *p* ≥ 0.05	Male		Female		Total	
	*n*	%	*n*	%	*N*	%
0–199 cells/μL	58	34.73	14	8.38	72	43.11
200–499 cells/μL	52	31.14	11	6.59	63	37.72
>500 cells/μL	19	11.38	13	7.78	32	19.16

**Table 5 medicina-59-00440-t005:** Distribution of study cases by CDC, HIV/AIDS classification.

Classification of the Studied Cases	n	%
Stage 1	32	19.16
Stage 2	51	30.54
Stage 3	84	50.30

**Table 6 medicina-59-00440-t006:** Distribution of cases based on sex and metabolic syndrome and liver enzymes.

Laboratory Marker	Value	Male		Female		Total	
		*n*	%	*N*	%	*N*	%
ALT	normal	78	60.47	31	82	109	65.27
	abnormal	51	39.53	7	18	58	34.73
AST	normal	65	50.39	28	74	93	55.69
	abnormal	64	49.61	10	26	74	44.31
GGT	normal	65	50.39	32	84	97	58.08
	abnormal	64	49.61	6	16	70	41.92
Cholesterol	normal	89	68.99	25	66	114	68.26
	abnormal	40	31.01	13	34	53	31.74
HDL-COL	normal	123	95.35	35	92	158	94.61
	abnormal	6	4.65	3	8	9	5.39
LDL-COL	normal	103	79.84	31	82	134	80.24
	abnormal	26	20.16	7	18	33	19.76
Triglycerides	normal	72	55.81	23	61	95	56.89
	abnormal	57	44.19	15	39	72	43.11

**Table 7 medicina-59-00440-t007:** Distribution of opportunistic infections by CDC stage in our study group.

	Stage 1		Stage 2		Stage 3		Total	
HIV/AIDS Status	*n*	%	*n*	%	*N*	%	*N*	%
No opportunistic infections	18	10.78	15	8.98	24	14.37	57	34.13
Opportunistic infections	13	7.78	43	25.75	54	32.34	110	65.87

**Table 8 medicina-59-00440-t008:** Distribution of new HIV/AIDS cases by co-infections.

Co-Infections	Men		Women		Total	
*p <* 0.05	*n*	%	*n*	%	*n*	%
HBV	15	8.98	14	8.38	29	17.37
HCV	6	3.59	5	2.99	11	6.59
TB	8	4.79	1	0.60	9	5.39
Syphilis	1	0.60	2	1.20	3	1.80
Candidiasis	39	23.35	19	11.38	58	34.73
Toxoplasmosis	3	1.80	0	0.00	3	1.80
CMV	1	0.60	1	0.60	2	1.20
VZV	4	2.40	1	0.60	5	2.99
Herpesviruses	3	1.80	1	0.60	4	2.40
*Cryptococcus*	1	0.60	0	0.00	1	0.60
SARS CoV-2	6	3.59	8	4.79	14	8.38

**Table 9 medicina-59-00440-t009:** Distribution of new HIV/AIDS cases by ART regimen.

ART Regimen	*N*	%
BIC/FTC/TAF (Bictegravir/Emtricitabine/Tenofovir alafenamide)	56	33.53
DTG/3TC (Dolutegravir/Lamivudine)	38	22.75
DOR/3TC/TDF (Doravirine/Lamivudine/Tenofovir disoproxil)	33	19.76
DTG/ABC/3TC (Dolutegravir/Abacavir/Lamivudine)	5	2.99
Other	35	20.96

**Table 10 medicina-59-00440-t010:** Distribution by CD4+ T-lymphocyte level and sex, one month after ART.

CD4 Levels *p* ≤ 0.05	Male		Female		Total	
	*n*	%	*n*	%	*n*	%
0–200 cells/μL	47	28.14	9	5.39	56	33.53
200–499 cells/μL	59	35.33	15	8.98	74	44.31
>500 cells/μL	23	13.77	14	8.38	37	22.16

**Table 11 medicina-59-00440-t011:** Distribution by HIV viral load level, at initial assessment, and one month after ART.

HIV Viral Load (*p* < 0.05)	Initial Assessment		One Month after ART	
	*n*	%	*n*	%
Undetectable (<40 copies/mL)	9	5.39	85	50.9
Detectable >40 copies/mL	158	94.6	82	49.1

## Data Availability

All data generated or analyzed during this study are included in this published article.

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
