# Peer review of "Profile of Newly Diagnosed Patients with HIV Infection in North-Eastern Romania"

_medicina, 2023, doi:10.3390/medicina59030440_

Round 1

Reviewer 1 Report

see attached doc

Reviewer 2 Report

The paper is well composed, the title is well defined, to which the text of the paper and the results gave an answer. It is easy to follow, useful for readers, the topic is always actual.

The goal is not clearly stated in the line, and I believe that it must be defined.

The methodology is well conceived. However, it should be supplemented with more details about the methods used to identifie newly infected patients. They are concisely stated, it should be written which tests were used for serological diagnosis and serological confirmation, as well as which quantitative real time PCR was used for   identifying HIV viremia as well as monitoring the success of therapy. In this sense, to enrich the methodology.

The results are well presented, the tables are clear. If the data can be obtained, the mode of infection of the newly infected patients who are the observed cohort, should be added. If it is possible to get the data, show this result as well, compare whether there is a difference in the way of transmission by gender and  age , educational level, usual pattern of behavior.

In the discussion part, if the paper is enriched with results about the way of  transmission, discuss your result in the context of the already commented topic in one part of this chapter.

Supplement with references if the results and discussion are enriched.

Reviewer 3 Report

ABSTRACT

“we aimed to characterize the trends of newly diagnosed patients with HIV/AIDS, during 2021-2022”: I understand that the word “trend” is associated with  longitudinal studies, I think that “aspect” is better

“We reviewed the prevalence, and associated comorbidities of this patients in correspondence with national and global results”: I understand that the word “prevalence” is associated with population studies, I think that “frequence” is better.

INTRODUCTION

The introduction is very long. I think that same parts can be changed for other issues such as comorbidities and opportunistic diseases at the time HIV diagnosis.

METHODS AND RESULTS

I suggest using the load viral as log10 scale instead copies/ml. In this way, the Figure 4 will be more clear.

In my opinion, from line 99 to line 106 cannot be considered methods, as the results of these methods are not your own results.

The last paragraph give us the study objetive “we aimed to characterize the trends of newly diagnosed patients with HIV/AIDS, during 2021-2022 in Northeastern Romania. We reviewed the prevalence, and associated comorbidities of the newly diagnosed patients with HIV infection in 110 correspondence with national and global results”. I think this part should be placed in the last paragraph of the Introduction section.

It is necessary or even better to explain criteria to participants such as “having HIV infection” and “having AIDS-related symptoms”. Does the category “having AIDS-related symptoms” correlate with CDC stage 1? If this is true, the frequencies between Table 3 and Table 4 are not the same.

Sometimes is used “sex” sometimes is used “gender”, I suggest to use only “sex” since the categories are “male” and “female”.

Acronym rules revision is required, eg line 92 and line 159 (Centers for Disease Control and Prevention (CDC)). Table 8, please add the meaning of the ART regimen acronyms.

The part of lines 163 to 169 is not results, it corresponds to Methods section. Additionally, in this part, different reference values between males and females in HDL-COL were not considered.

I suggest to correlate the CDC stage to comorbidities and opportunistic diseases.  

DISCUSSION:

The paragraph between lines 211 and 213 is not necessary.

Paragraphs between lines 214 and 219 are from the introduction section and not the discussion section.

It is necessary to improve this section. The discussion section presents many studies from literature and compares them with the results of the your study. However, there is no real discution about the results of your study.

Round 2

Reviewer 3 Report

The expressions like "HIV-positive individuals" or "HIV infected cases"or "patients with HIV/AIDS" , "HIV-infected patients", should be replaced by ¨People living with HIV (PLWH)".

The p-value in the Table 11 is lesser than 0.05 not igual 0.5 since the values of "initial assessment"  and "One month after ART" is very different. 
